# Dynamic Non-monotone Submodular Maximization

**Kiarash Banihashem**[*][†]
kiarash@umd.edu
University of Maryland

**Leyla Biabani**[*][‡]
l.biabani@tue.nl
TU Eindhoven

**Samira Goudarzi**[*][†]
samirag@umd.edu
University of Maryland

**MohammadTaghi Hajiaghayi**[*][†]
hajiagha@cs.umd.edu
University of Maryland

**Peyman Jabbarzade**[*][†]
peymanj@umd.edu
University of Maryland

**Morteza Monemizadeh**[*][‡]
m.monemizadeh@tue.nl
TU Eindhoven

## Abstract

Maximizing submodular functions has been increasingly used in many applications of machine learning, such as data summarization, recommendation systems, and feature selection. Moreover, there has been a growing interest in both submodular maximization and dynamic algorithms. In 2020, Monemizadeh [46] and Lattanzi, Mitrovic, Norouzi-Fard, Tarnawski, and Zadimoghaddam [40] initiated developing dynamic algorithms for the monotone submodular maximization problem under the cardinality constraint $k$. In 2022, Chen and Peng [15] studied the complexity of this problem and raised an important open question: "*Can we extend [fully dynamic] results (algorithm or hardness) to non-monotone submodular maximization?*". We affirmatively answer their question by demonstrating a reduction from maximizing a non-monotone submodular function under the cardinality constraint $k$ to maximizing a monotone submodular function under the same constraint. Through this reduction, we obtain the first dynamic algorithms solving the non-monotone submodular maximization problem under the cardinality constraint $k$. We've derived two algorithms, both maintaining an $(8 + \epsilon)$-approximate of the solution. The first algorithm requires $\mathcal{O}(\epsilon^{-3}k^3 \log^3(n) \log(k))$ oracle queries per update, while the second one requires $\mathcal{O}(\epsilon^{-1}k^2 \log^3(k))$. Furthermore, we showcase the benefits of our dynamic algorithm for video summarization and max-cut problems on several real-world data sets.

## 1  Introduction

Submodular functions are powerful tools for solving real-world problems as they provide a theoretical framework for modeling the famous "*diminishing returns*" [30] phenomenon arising in a variety of practical settings. Many theoretical problems such as those involving graph cuts, entropy-based clustering, coverage functions, and mutual information can be cast in the submodular maximization framework. As a result, submodular functions have been increasingly used in many applications of machine learning such as data summarization [52, 51, 50], feature selection [17, 19, 18, 38], and recommendation systems [24]. These applications include both the monotone and non-monotone versions of the maximization of submodular functions.

**Applications of non-monotone submodular maximization.** The general problem of non-monotone submodular maximization has been studied extensively in [27, 12, 11, 43, 5, 47]. This

---

[*]equal contribution

[†]Department of Computer Science, University of Maryland, College Park, MD, USA.

[‡]Department of Mathematics and Computer Science, Eindhoven University of Technology, the Netherlands.

37th Conference on Neural Information Processing Systems (NeurIPS 2023).

problem has numerous applications in video summarization, movie recommendation [43], and revenue maximization in viral marketing [35][4]. An important application of this problem appears in maximizing the difference between a monotone submodular function and a linear function that penalizes the addition of more elements to the set (e.g., the coverage and diversity trade-off). An illustrative example of this application is the maximum facility location in which we want to open a subset of facilities and maximize the total profit from served clients plus the cost of facilities we did not open [21]. Another important application occurs when expressing learning problems such as feature selection using weakly submodular functions [17, 38, 25, 49].

**Our contribution.** In this paper, we consider the non-monotone submodular maximization problem under cardinality constraint $k$ in the *fully dynamic setting*. In this model, we have a *universal ground set* $V$. At any time $t$, ground set $V_t \subseteq V$ is the set of elements that are inserted but not deleted after their last insertion till time $t$. More formally, we assume that there is a sequence of "updates" such that each update either inserts an element to $V_{t-1}$ or deletes an element from $V_{t-1}$ to form $V_t$.

We assume that there is a (non-monotone) submodular function $f$ that is defined over the universal ground set $V$. Our goal is to maintain, at each point in time, a set of size at most $k$ whose submodular value is maximum among any subset of $V_t$ of size at most $k$.

Since calculating such a set is known to be NP-hard [27] even in the offline setting (where you get all the items at the same time), we focus on providing algorithms with provable approximation guarantees, while maintaining fast update time. This is challenging as elements may be inserted or deleted, possibly in an adversarial order. While several dynamic algorithms exist for monotone submodular maximization, non-monotone submodular maximization is a considerably more challenging problem as adding elements to a set may decrease its value.

In STOC 2022, Chen and Peng [15] raised the following open question:

**Open problem:** "Can we extend [fully dynamic] results (algorithm or hardness) to non-monotone submodular maximization?"

In this paper, we answer their question affirmatively by providing the first dynamic algorithms for non-monotone submodular maximization.

To emphasize the significance of our result, it should be considered that although monotone submodular maximization under cardinality constraint has a tight $\frac{e}{e-1}$ approximation algorithm in the offline mode and nearly tight $(2 + e)$ approximation algorithms for both streaming and dynamic settings, there is a hardness result for the non-monotone version stating that it is impossible to obtain a 2.04 (i.e., 0.491) approximation algorithm for this problem even in the offline setting[31], and to the best of our knowledge, the current state of the art algorithms for this problem have 2.6 and 3.6 (i.e., 0.385 and 0.2779) approximation guarantees for offline [10] and streaming settings [2], respectively.

We obtain our result, by proposing a general reduction from the problem of dynamically maintaining a non-monotone submodular function under cardinality constraint $k$ to developing a dynamic thresholding algorithm for maximizing monotone submodular functions under the same constraint. We first define $\tau$-thresholding dynamic algorithms that we use in our reduction.

**Definition 1.1** ($\tau$-Thresholding Dynamic Algorithm). Let $\tau > 0$ be a parameter. We say a dynamic algorithm is $\tau$-thresholding if at any time $t$ of sequence $\Xi$, it reports a set $S_t \subseteq V_t$ of size at most $k$ such that

- **Property 1:** either **a)** $S_t$ has $k$ elements and $f(S_t) \geq k\tau$, or **b)** $S_t$ has less than $k$ elements and for any $v \in V_t \setminus S_t$, the marginal gain $\Delta(v|S_t) < \tau$.

- **Property 2:** The number of elements changed in any update, i.e, $|S_{t+1} \setminus S_t| + |S_t \setminus S_{t+1}|$, is not more than the number of queries made by the algorithm during the update.

In the above definition, the first property reflects the main intuition of threshold-based algorithms, while the last property is a technical condition required in our analysis. It's worth noting that the thresholding technique has been used widely for optimizing submodular functions [46, 41, 26, 40, 16]. We next state our main result, which is a general reduction.

---

[4]The problem of selecting a subset of people in a social network to maximize their influence in a viral marketing campaign can be modeled as a constrained submodular maximization problem. When we introduce a cost, then the influence minus the cost is modeled as non-monotone submodular maximization problems [9, 8, 34].

**Theorem 1.2** (**Reduction Metatheorem**). *Suppose that $f : 2^V \to \mathbb{R}_{\geq 0}$ is a (possibly non-monotone) submodular function defined on subsets of a ground set $V$ and let $k \in \mathbb{N}$ be a parameter.*

*Assume that for any given value of $\tau$, there exists a $\tau$-thresholding dynamic algorithm with an expected (amortized) $O(g(n,k))$ oracle queries per update. Then, there exist the following dynamic algorithms:*

- *A dynamic algorithm with an approximation guarantee of $(8 + \varepsilon)$ using an expected (amortized) $O(k + \min(k, g(n,k)) \cdot g(n,k) \cdot \varepsilon^{-1} \log(k))$ oracle queries per update.*

- *A dynamic algorithm maintaining a $(10 + \varepsilon)$-approximate solution of the optimal value of $f$ using an expected (amortized) $O(\min(k, g(n,k)) \cdot g(n,k) \cdot \varepsilon^{-1} \log(k))$ oracle calls per update.*

In [46], Monemizadeh developed a dynamic algorithm for monotone submodular maximization under cardinality constraint $k$, which requires an amortized $O(\varepsilon^{-2} k^2 \log^3(n))$ number of oracle queries per update. Interestingly, in the appendix, we show that this algorithm is indeed $\tau$-thresholding (for any given $\tau$). Now, if we use this $\tau$-thresholding dynamic algorithm inside our reduction Metatheorem 1.2, we obtain a dynamic algorithm that maintains a $(8 + \varepsilon)$-approximate solution using an expected amortized $O(\varepsilon^{-3} k^3 \log^3(n) \log(k))$ oracle queries per update.

The recent paper [6] of Banihashem, Biabani, Goudarzi, Hajiaghayi, Jabbarzade, and Monemizadeh develops a new dynamic algorithm for monotone submodular maximization under cardinality constraint $k$, which uses an expected $O(\varepsilon^{-1} k \log^2(k))$ number of oracle queries per update. A similar proof shows that this new algorithm is $\tau$-thresholding as well. We have provided its pseudocode and a detailed explanation on why this algorithm is indeed $\tau$-thresholding in the appendix. By exploiting this algorithm in our Reduction Metatheorem 1.2, we can reduce the number of oracle queries mentioned to an expected number of $O(\varepsilon^{-2} k^2 \log^3(k))$ per update.

The second result in Theorem 1.2 is also of interest as it can be used to devise a dynamic algorithm for non-monotone submodular maximization with polylogarithmic query complexity if one can provide a $\tau$-thresholding dynamic algorithm for maximizing monotone submodular functions (under the cardinality constraint $k$) with polylogarithmic query complexity.

## 1.1 Preliminaries

**Submodular maximization.** Let $V$ be a ground set of elements. We say a function $f : 2^V \to \mathbb{R}_{\geq 0}$ is a *submodular* function if for any $A, B \subseteq V$, $f(A) + f(B) \geq f(A \cup B) + f(A \cap B)$. Equivalently, $f$ is a submodular function if for any subsets $A \subseteq B \subseteq V$ and for any element $e \in V \setminus B$, it holds that $f(A \cup \{e\}) - f(A) \geq f(B \cup \{e\}) - f(B)$ . We define $\Delta(e|A) := f(A \cup \{e\}) - f(A)$ the *marginal gain* of adding the element $e$ to set $A$. Similarly, we define $\Delta(B|A) := f(A \cup B) - f(A)$ for any sets $A, B \subseteq V$. Function $f$ is *monotone* if $f(A) \leq f(B)$ holds for any $A \subseteq B \subseteq V$, and it is *non-monotone* if it is not necessarily the case. In the submodular maximization problem under cardinality constraint $k$, we seek to compute a set $S^*$ such that $|S^*| \leq k$ and $f(S^*) = \max_{|S| \leq k} f(S)$, where $f$ is a submodular function and $k \in \mathbb{N}$ is a given parameter.

**Query access model.** Similar to recent dynamic works [40, 15], we assume the access to a submodular function $f$ is given by an *oracle*. The oracle allows *set queries* such that for every subset $A \subseteq V$, one can query the value $f(A)$. In this query access model, the marginal gain $\Delta_f(e|A) \doteq f(A \cup \{e\}) - f(A)$ for every subset $A \subseteq V$ and an element $e \in V \setminus A$, can be computed using two set queries. To do so, we first query $f(A \cup \{e\})$ and then $f(A)$.

**Dynamic model.** Let $\Xi$ be a sequence of inserts and deletes of an underlying universe $V$. We assume that $f : 2^V \to \mathbb{R}_{\geq 0}$ is a (possibly non-monotone) submodular function defined on subsets of the universe $V$. We define time $t$ to be the $t^{\text{th}}$ update (i.e., insertion or deletion) of sequence $\Xi$. We let $\Xi_t$ be the sub-sequence of updates from the beginning of sequence $\Xi$ till time $t$ and denote by $V_t \subseteq V$ the set of elements that are inserted but not deleted from the beginning of the sequence $\Xi$ till any time $t$. That is, $V_t$ is the current ground set of elements. We let $OPT_t = \max_{S \subseteq V_t : |S| \leq k} f(S)$.

**Query complexity.** The *query complexity* of a dynamic $\alpha$-approximate algorithm is the number of oracle queries that the algorithm must make to compute a solution $S_t$ with respect to ground set $V_t$

whose submodular value is an $\alpha$-approximation of $OPT_t$. That is, $|S_t| \leq k$ and $f(S_t) \geq \alpha \cdot OPT_t$. Observe that the dynamic algorithm remembers every query it has made so far. Thus results of queries made in previous times may help find $S_t$ in current time $t$.

**Oblivious adversarial model.** The dynamic algorithms that we develop in this paper are in the *oblivious adversarial model* as is common for analysis of randomized data structures such as universal hashing [13]. The model allows the adversary, who is aware of the submodular function $f$ and the algorithm that is going to be used, to determine all the arrivals and departures of the elements in the ground set $V$. However, the adversary is unaware of the random bits used in the algorithm and so cannot choose updates adaptively in response to the randomly guided choices of the algorithm. Equivalently, we can suppose that the adversary prepares the full input (insertions and deletions) before the algorithm runs.

## 1.2 Related Work

**Offline algorithms.** The offline version of non-monotone submodular maximization was first studied by Feige, Mirrokni, and Vondrák in [27]. They studied *unconstrained non-monotone submodular maximization* and developed constant-factor approximation algorithms for this problem. In the offline query access model, they showed that a subset $S$ chosen uniformly at random has a submodular value which is a 4-approximation of the optimal value for this problem. In addition, they also described two local search algorithms. The first uses $f$ as the objective function, and provides 3-approximation and the second uses a noisy version of $f$ as the objective function and achieves an improved approximation guarantee 2.5 for maximizing unconstrained non-monotone non-negative submodular functions. Interestingly, they showed $(2 - \varepsilon)$-approximation for symmetric submodular functions would require an exponential number of queries for any fixed $\varepsilon > 0$.

Oveis Gharan and Vondrák [32] showed that an extension of the 2.5-approximation algorithm can be seen as *simulated annealing* method which provides an improved approximation of roughly 2.4. Later, Buchbinder, Feldman, Naor, and Schwartz [11] at FOCS'12, presented a randomized linear time algorithm achieving a tight approximation guarantee of 2 that matches the known hardness result of [27]. Bateni, Hajiaghayi, and Zadimoghaddam [9, 8] and Gupta, Roth, Schoenebeck, and Talwar [34] independently studied non-monotone submodular maximization subject to cardinality constraint $k$ in the offline and secretary settings. In particular, Gupta *et al.* [34] obtained an offline 6.5-approximation for this problem.

All of the aforementioned approximation algorithms are offline, where the whole input is given in the beginning, whereas the need for real-time analysis of rapidly changing data streams motivates the study of this problem in settings such as the dynamic model that we study in this paper.

**Streaming algorithms.** The dynamic model that we study in this paper is closely related to the streaming model [3, 36]. However, the difference between these two models is that in the streaming model, we maintain a data structure using which we compute a solution at the end of the stream and so, the time to extract the solution is not important as we do it once. However, in the dynamic model, we need to maintain a solution after every update, thus, the update time of a dynamic algorithm should be as fast as possible.

The known streaming algorithms [44, 28, 29] work in the insertion-only streaming model and they do not support deletions as well as insertions. Indeed, there are streaming algorithms [37, 45] for the monotone submodular maximization problem that support deletions, but the space and the update time of these algorithms depend on the number of deletions which could be $\Omega(n)$, where $n = |V|$ is the size of ground set $V$.

For monotone submodular maximization, Badanidiyuru, Mirzasoleiman, Karbasi, and Krause [4] proposed an insertion-only streaming algorithm with a $(2+\varepsilon)$-approximation guarantee under a cardinality constraint $k$. Chekuri, Gupta, and Quanrud [14] presented (insertion-only) streaming algorithms for maximizing monotone and non-monotone submodular functions subject to $p$-matchoid constraint[5]. Later, Mirzasoleiman, Jegelka, and Krause [44] and Feldman, Karbasi, and Kazemi [28] devel-

---

[5]For non-monotone submodular maximization subject to cardinality constraint $k$, Chekuri, Gupta, and Quanrud [14] claimed that they obtained 4.7-approximation algorithm. However, Alaluf, Ene, Feldman, Nguyen, and Suh [1] found an error in the proof of this approximation guarantee.

oped streaming algorithms with better approximation guarantees for maximizing a non-monotone function under a $p$-matchoid constraint. Currently, the best streaming algorithm for maximizing a non-monotone submodular function subject to a cardinality constraint is due to Alaluf, Ene, Feldman, Nguyen, and Suh [1] whose approximation guarantee is $3.6 + \varepsilon$, improving the $5.8$-approximation guarantee that was proposed by Feldman *et al.* [28].

**Dynamic algorithms.**  At NeurIPS 2020, Lattanzi, Mitrovic, Norouzi-Fard, Tarnawski, and Zadi-moghaddam [40] and Monemizadeh [46] initiated the study of submodular maximization in the dynamic model. They presented dynamic algorithms that maintain $(2 + \varepsilon)$-approximate solutions for maximizing a monotone submodular function subject to cardinality constraint $k$. Later, at STOC 2022, Chen and Peng [15] studied the complexity of this problem and they proved that developing a $c$-approximation dynamic algorithm for $c < 2$ is not possible unless we use a number of oracle queries polynomial in the size of ground set $V$. In 2023, Banihashem, Biabani, Goudarzi, Hajiaghayi, Jabbarzade, and Monemizadeh[7] developed an algorithm for monotone submodular maximization problem under cardinality constraint $k$ using a polylogarithmic amortized update time. Concurrent works of Banihashem, Biabani, Goudarzi, Hajiaghayi, Jabbarzade, and Monemizadeh[6] and Duet-ting, Fusco, Lattanzi, Norouzi-Fard, and Zadimoghaddam[22] developed the first dynamic algorithms for monotone submodular maximization under a matroid constraint. Authors of [6] also improve the algorithm of [46] for monotone submodular maximization subject to cardinality constraint $k$. There are also studies on the dynamic model of influence maximization, which shares similarities with submodular maximization [48].

In this paper, for the first time, we study the generalized version of their problem by presenting an algorithm for maximizing the non-monotone submodular functions in the dynamic setting.

## 2   Dynamic algorithm

In this section, we explain the algorithm that we use in the reduction that we stated in Metatheorem 1.2. The pseudocode of our algorithm is given in Algorithm 1, Algorithm 2, and Algorithm 3.

Such reductions were previously proposed in the offline model by [34], and later works extended this idea to the streaming model [14, 44]. We develop a reduction in the dynamic model inspired by these works, though in our proof, we require a tighter analysis to obtain the approximation guarantee in our setting.

We consider an arbitrary time $t$ of sequence $\Xi$ where $V_t$ is the set of elements inserted before time $t$, but not deleted after their last insertion. Let $OPT_t^* = \max_{S \subseteq V_t : |S| \le k} f(S)$. For simplicity, we drop $t$ from $V_t$ and $OPT_t^*$, when it is clear from the context.

In the following, we assume that the value of $OPT$ is known. Although the exact value of $OPT^*$ is unknown, we can maintain parallel runs of our dynamic algorithm for different guesses of the optimal value. By using $(1 + \varepsilon')^i$, where $i \in \mathbb{Z}$ as our guesses for the optimal value, one of our guesses $(1 + \varepsilon')$-approximates the value of $OPT^*$. We show that the output of our algorithm satisfies the approximation guarantee in the run whose $OPT$ $(1 + \varepsilon')$-approximates the value of $OPT^*$. Later, in the appendix, we show that it is enough to consider each element $e$ only in runs $i$ for which we have $\frac{\varepsilon'}{k} \cdot (1 + \varepsilon')^i \le f(e) \le (1 + \varepsilon')^i$. This method increases the query complexity of our dynamic algorithm by only a factor of $O(\varepsilon^{-1} \log k)$.

Our approach for solving the non-monotone submodular maximization is to first run the thresholding algorithm with input set $V$ to find a set $S_1$ of at most $k$ elements. Since $f$ is non-monotone, subsets of $S_1$ might have a higher submodular value than $f(S_1)$. Then, we use an $\alpha$-approximation algorithm (for $0 < \alpha \le 1$) to choose a set $S_1' \subseteq S_1$ with guarantee $\mathbf{E}[f(S_1')] \ge \alpha \cdot \max_{C \subseteq S_1} f(C)$. Next, we run the thresholding algorithm with the input set $V \backslash S_1$ and compute a set $S_2$. At the end, we return set $S = \arg\max_{C \in \{S_1, S_1', S_2\}} f(C)$. Intuitively, for an optimal solution $S^*$, if $f(S_1 \cap S^*)$ is a good approximation of $OPT$, then $f(S_1')$ is a good approximation of $OPT$. On the other hand, if both $f(S_1)$ and $f(S_1 \cap S^*)$ are small with respect to $OPT$, then we can ignore the elements of $S_1$ and show that we can find a set $S_2 \subseteq V \setminus S_1$ of size at most $k$ whose submodular value is a good approximation of $OPT$. The following lemma proves that the submodular value of $S$ is a reliable approximation of the optimal solution. The formal proof of this lemma can be found in Section 2.

**Lemma 2.1** (Approximation Guarantee). *Assuming that $OPT^* \in [\frac{OPT}{1+\varepsilon'}, OPT]$, the expected submodular value of set $S$ is $\mathbf{E}[f(S)] \geq (1 - O(\varepsilon'))\frac{OPT^*}{6+\frac{1}{\alpha}}$.*

Next, we explain the steps of our reduction in detail.

Let us first fix the threshold $\tau = \frac{OPT}{k(3+1/(2\alpha))}$. Then, we fix a $\tau$-thresholding dynamic algorithm (for example, [46] or [6]) and suppose we denote it by DYNAMICTHRESHOLDING. Before sequence $\Xi$ of updates starts, we create two independent instances $\mathcal{I}_1$ and $\mathcal{I}_2$ of DYNAMICTHRESHOLDING. The first instance will maintain set $S_1$ and the second instance will maintain set $S_2$. For instance $\mathcal{I}_i$ where $i \in \{1, 2\}$, we consider the following subroutines:

- INSERT$_{\mathcal{I}_i}(v)$: This subroutine inserts an element $v$ to instance $\mathcal{I}_i$.

- DELETE$_{\mathcal{I}_i}(v)$: Invoking this subroutine will delete the element $v$ from instance $\mathcal{I}_i$.

- EXTRACT$_{\mathcal{I}_i}$: This subroutine returns the maintained set (of size at most $k$) of $\mathcal{I}_i$.

**Extracting $S_1$.** After the update at time $t$, first, we would like to set $Z = S_1^- \cup \{v\}$ or $Z = S_1^- \setminus \{v\}$, if the update is the insertion of an element $v$ or the deletion of an element $v$, respectively, where $S_1^-$ is the set $S_1$ that instance $\mathcal{I}_1$ maintains just before this update. To find set $S_1^-$, we just need to invoke subroutine EXTRACT$_{\mathcal{I}_1}$. If the update is an insertion, we insert it into instance $\mathcal{I}_1$ using INSERT$_{\mathcal{I}_1}(v, \tau)$, and if the update is a deletion, we delete $v$ from both $\mathcal{I}_1$ and $\mathcal{I}_2$ using DELETE$_{\mathcal{I}_1}(v)$ and DELETE$_{\mathcal{I}_2}(v)$. We then invoke EXTRACT$_{\mathcal{I}_1}$ once again to return set $S_1$.

**Extracting $S_1'$.** Buchbinder *et al.* [11] developed a method to extract a subset $S_1' \subseteq S_1$ whose submodular value is a good approximation of $\max_{C \subseteq S_1} f(C)$. In this algorithm, we start with two solutions $\emptyset$ and $S_1$. The algorithm considers the elements (in arbitrary order) one at a time. For each element, it determines whether it should be added to the first solution or removed from the second solution. Thus, after a single pass over set $S_1$, both solutions completely coincide, which is the solution that the algorithm outputs. They show that a (deterministic) greedy choice in each step obtains 3-approximation of the best solution in $S_1$. However, if we combine this greedy choice with randomization, we can obtain a 2-approximate solution. Since we do a single pass over set $S_1$, the number of oracle queries is $O(|S_1|)$.

The second algorithm that we can use to extract $S_1'$ is a random sampling algorithm proposed by Feige *et al.* [27], which choose every element in $S_1$ with probability $1/2$. They show that this random sampling returns a set $S_1'$ whose approximation factor is $1/4$ of $\max_{C \subseteq S_1} f(C)$, and its number of oracle calls is $O(1)$. We denote either of these two methods by SUBSETSELECTION.

**Extracting $S_2$.** Next, we would like to update the set $S_2$ that is maintained by instance $\mathcal{I}_2$. To do this, for every element $u \in Z \setminus S_1$, we add it to $\mathcal{I}_2$ using INSERT$_{\mathcal{I}_2}(u, \tau)$, and for every element $u \in S_1 \setminus Z$, we delete it from $\mathcal{I}_2$ using DELETE$_{\mathcal{I}_2}(u, \tau)$. Finally, when $\mathcal{I}_2$ exactly includes all the current elements other than the ones in $S_1$, we call subroutine EXTRACT$_{\mathcal{I}_2}$ to return set $S_2$.

**Corollary 2.2.** *We obtain the $(8 + \varepsilon)$ approximation guaranty stated in the Metatheorem 1.2 by using the local search method for our SUBSETSELECTION, and we get the $(10 + \varepsilon)$ approximation guaranty by utilizing the random sampling method for our SUBSETSELECTION subroutine.*

*Proof.* These are immediate results of Lemma 2.1, and $\alpha$ being $\frac{1}{2}$ and $\frac{1}{4}$ in the local search method and random sampling method, respectively. □

---

**Algorithm 1** Initialization$(k, OPT)$

---

1: $\tau \leftarrow \frac{OPT}{k(3+1/(2\alpha))}$, where $\alpha$ is $\frac{1}{2}$ or $\frac{1}{4}$ based on the selection of algorithm for SUBSETSELECTION.
2: Instantiate two independent instances $\mathcal{I}_1$ and $\mathcal{I}_2$ of DYNAMICTHRESHOLDING for monotone submodular maximization under cardinality constraint $k$ using $\tau$

---

---

**Algorithm 2** UPDATE($v$)

1: $Z \leftarrow$ EXTRACT$_{\mathcal{I}_1}$
2: **if** UPDATE($v$) is an insertion **then**
3:     Invoke INSERT$_{\mathcal{I}_1}(v)$, $Z \leftarrow Z \cup \{v\}$
4: **else**
5:     Invoke DELETE$_{\mathcal{I}_1}(v)$, DELETE$_{\mathcal{I}_2}(v)$, $Z \leftarrow Z \setminus \{v\}$
6: $S_1 \leftarrow$ EXTRACT$_{\mathcal{I}_1}$
7: $S_1' \leftarrow$ SUBSETSELECTION($S_1$)
8: **for** $u \in S_1 \setminus Z$ **do**
9:     DELETE$_{\mathcal{I}_2}(u)$
10: **for** $u \in Z \setminus S_1$ **do**
11:     INSERT$_{\mathcal{I}_2}(u)$
12: $S_2 \leftarrow$ EXTRACT$_{\mathcal{I}_2}$
13: Return $\arg\max_{C \in \{S_1, S_1', S_2\}} f(C)$

---

**Algorithm 3** SUBSETSELECTION($S$)

1: **function** UNIFORMSUBSET($S$)
2:     $T \leftarrow \emptyset$
3:     **for** $s \in S$ **do**
4:         **if** $Coin(\frac{1}{2})$ **then**                           $\triangleright$ With probability $\frac{1}{2}$
5:             $T \leftarrow T \cup \{s\}$
6:     **return** $T$
7: **function** LOCALSEARCHSUBSET($S$)
8:     $X_0 \leftarrow \emptyset$, $Y_0 \leftarrow S$.
9:     **for** $i = 1$ to $|S|$ **do**
10:         $a_i \leftarrow f(X_{i-1} \cup \{s_i\}) - f(X_{i-1})$, $b_i \leftarrow f(Y_{i-1} \setminus \{s_i\}) - f(Y_{i-1})$
11:         $a_i' \leftarrow \max(a_i, 0)$, $b_i' \leftarrow \max(b_i, 0)$
12:         **if** $a_i' = b_i' = 0$ **then** $a_i'/(a_i' + b_i') = 0$
13:         **with probability** $a_i'/(a_i' + b_i')$ **do:**
14:         $X_i \leftarrow X_{i-1} \cup \{s_i\}$, $Y_i \leftarrow Y_{i-1}$
15:         **else do:** $X_i \leftarrow X_{i-1}$, $Y_i \leftarrow Y_{i-1} \setminus \{s_i\}$
16:     **return** $X_{|S|}$ (or equivalently $Y_{|S|}$)

---

**Analysis.** In this section, we prove the correctness of our algorithms and analyze the number of oracle queries of our algorithms, which finishes the proof of Theorems 1.2.

Consider an arbitrary time $t$. Let $S_t$ be the reported set of DYNAMICTHRESHOLDING at time $t$. Recall that $V_t$ is the ground set at time $t$, and we drop the $t$ for simplicity, so we use $V$ and $S$ to denote $V_t$ and $S_t$. We first present Lemma 2.3 whose proof is given in the appendix. Then we proceed to prove Lemma 2.1 and Theorem 1.2

**Lemma 2.3.** *Suppose that set $S$ satisfies Property 1.b of Definition 1.1. It means that $S$ has less than $k$ elements and for any $v \in V \setminus S$, the marginal gain $\Delta(v|S) < \tau$. Then, for any arbitrary subset $C \subseteq V$, we have $f(S) \geq f(S \cup C) - |C| \cdot \tau$.*

**Proof of Lemma 2.1:** Assume that at a fixed time $t$, $OPT^*$ and $S^*$ are the optimal value and an optimal solution for the submodular maximization of function $f$ under cardinality constraint $k$. This means that $|S^*| \leq k$ and $f(S^*) = OPT^*$. Recall that $\tau = \frac{OPT}{k(3 + \frac{1}{2\alpha})}$, where $OPT$ is our guess for the optimal value. Also, by assumption we have $OPT^* \in [\frac{OPT}{1+\epsilon'}, OPT]$, or equivalently $OPT \in [OPT^*, (1 + \epsilon')OPT^*]$.

To prove the lemma, we claim that $\max(\mathbf{E}[f(S_1)], \mathbf{E}[f(S_1')], \mathbf{E}[f(S_2)]) \geq (1 - O(\varepsilon'))\frac{OPT^*}{6 + \frac{1}{\alpha}}$.

Suppose this claim is true. Using Jensen's inequality [23], we have $\mathbf{E}[\max(f(S_1), f(S_1'), f(S_2))] \geq \max(\mathbf{E}[f(S_1)], \mathbf{E}[f(S_1')], \mathbf{E}[f(S_2)])$, which yields $\mathbf{E}[f(S)] \geq (1 - O(\varepsilon'))\frac{OPT^*}{6 + 1/\alpha}$.

To prove the claim, we consider two cases. The first case is when $f(S_1 \cap S^*) \geq \frac{\tau k}{2\alpha}$ and the second case is if $f(S_1 \cap S^*) < \frac{\tau k}{2\alpha}$.

Suppose the first case is true. Then, the subset selection algorithm (either random sampling method or local search) returns $S_1'$ for which $\mathbf{E}[f(S_1')] \geq \alpha \cdot \max_{S \subseteq S_1} f(S)$. Since $S_1 \cap S^* \subseteq S_1$, we have

For the latter case, we show that $\mathbf{E}[f(S_1)] + \mathbf{E}[f(S_2)] \geq (1 - O(\varepsilon'))\frac{OPT^*}{3+1/2\alpha}$, inferring $\max\left(\mathbf{E}[f(S_1)], \mathbf{E}[f(S_2)]\right) \geq (1 - O(\varepsilon'))\frac{OPT^*}{6+1/\alpha}$. Indeed, since $S_1$ and $S_2$ are reported by an $\tau$-thresholding algorithm, if $|S_1| = k$ or $|S_2| = k$, then $\max(\mathbf{E}[f(S_1)], \mathbf{E}[f(S_2)])$ is at least $\tau k = \frac{OPT}{3+1/2\alpha}$ by the first property of $\tau$-thresholding algorithms.

Now suppose that $|S_1|, |S_2| < k$, which means that Property 1.b of Definition 1.1 holds for $S_1$ and $S_2$. Therefore, we have $f(S_1) \geq f(S_1 \cup S^*) - \tau|S^*|$ and $f(S_2) \geq f(S_2 \cup (S^* \setminus S_1)) - \tau|S^* \setminus S_1|$ by Lemma 2.3. Besides, we have $f(S_1 \cap S^*) - \frac{\tau k}{2\alpha} < 0$. Therefore,

$$f(S_1) + f(S_2) \geq f(S_1 \cup S^*) - \tau|S^*| + f(S_2 \cup (S^* \setminus S_1)) - \tau|S^* \setminus S_1| + f(S_1 \cap S^*) - \frac{\tau k}{2\alpha} \ .$$

Since $|S^* \setminus S_1| \leq |S^*| \leq k$ we have

$$f(S_1) + f(S_2) \geq f(S_1 \cup S^*) + f(S_2 \cup (S^* \setminus S_1)) + f(S_1 \cap S^*) - (2 + 1/(2\alpha))\tau k \ .$$

Since $S_1 \cap S_2 = \emptyset$ and $f$ is submodular, we have $f(S_1 \cup S^*) + f(S_2 \cup (S^* \setminus S_1)) \geq f(S_1 \cup S_2 \cup S^*) + f(S^* \setminus S_1)$. Additionally, by the submodularity and non-negativity of $f$, we have $f(S_1 \cap S^*) \geq f(S^*) - f(S^* \setminus S_1)$, because $f(S^* \setminus S_1) + f(S_1 \cap S^*) \geq f(S^*) + f(\emptyset)$. By adding the last two inequalities and using the non-negativity of $f$ once again, we get $f(S_1 \cup S^*) + f(S_2 \cup (S^* \setminus S_1)) + f(S_1 \cap S^*) \geq f(S_1 \cup S_2 \cup S^*) + f(S^*) \geq f(S^*) = OPT^*$. By putting everything together we have,

$$f(S_1) + f(S_2) \geq OPT^* - (2 + 1/(2\alpha))\tau k = OPT^* - (\frac{4\alpha + 1}{2\alpha})(\frac{OPT(2\alpha)}{6\alpha + 1}).$$

By using the assumption that $OPT \leq (1 + \epsilon')OPT^*$, we have,

$$f(S_1) + f(S_2) \geq OPT^*(1 - (\frac{(4\alpha + 1)(1 + \epsilon')}{6\alpha + 1})) \geq OPT^*(\frac{2\alpha - \epsilon'(4\alpha + 1)}{6\alpha + 1}) = (1 - O(\varepsilon'))\frac{OPT^*}{3 + 1/(2\alpha)}.$$

$\square$

**Proof of Theorem 1.2:**

As previously discussed, we've established in Lemma 2.1 and Corollary 2.2 that utilizing the local search method for the SUBSETSELECTION subroutine results in an approximation ratio of $(8 + \varepsilon)$, whereas the random sampling method achieves an approximation ratio of $(10 + \varepsilon)$. Thus, the only remaining aspect to address in proving this theorem is proving the query complexity of our proposed algorithm. In Lemma 2.4, we bound the number of queries made in each run of our algorithm per update, proving the bounds given in Theorem 1.2 by considering the extra $O(\varepsilon^{-1} \log k)$-factor caused by our parallel runs. $\square$

**Lemma 2.4.** *Let the random variable $Q_t$ denote the number of oracle calls that our algorithm in Theorem 1.2 makes at time $t$ in each of the parallel runs. Depending on whether the expected or expected amortized number of oracle calls made by the thresholding algorithm* DYNAMICTHRESHOLD *per each update is $O(g(n,k))$, if we choose the local search method as our* SUBSETSELECTION *subroutine, we have*

$$\mathbf{E}[Q_t] \in O(\min(k \cdot g(n,k), g(n,k)^2)) + O(k) \ ,$$

*or*

$$\mathbf{E}[\sum_{t=1}^{T} Q_t] \in O(T \cdot \min(k \cdot g(n,k), g(n,k)^2)) + O(k) \ ,$$

*and if we choose the random sampling method as our* SUBSETSELECTION *subroutine, we have*

$$\mathbf{E}[Q_t] \in O(\min(k \cdot g(n,k), g(n,k)^2)) \ ,$$

*or*

$$\mathbf{E}[\sum_{t=1}^{T} Q_t] \in O(T \cdot \min(k \cdot g(n,k), g(n,k)^2))$$

*Proof.* Consider the case where the expected number of oracle calls made by the thresholding algorithm DYNAMICTHRESHOLD per each update is $O(g(n,k))$. Per each update, our algorithm makes an update in instance $\mathcal{I}_1$ causing $O(g(n,k))$ oracle queries. Next, we make either $O(k)$ or 0 oracle queries for the SUBSETSELECTION subroutine, depending on the used method. We also make a series of updates in instance $\mathcal{I}_2$, each causing $O(g(n,k))$ oracle queries. The number of such updates is bounded by the number of changes in the output of instance $\mathcal{I}_1$, which is bounded by both $k$ and $O(g(n,k))$ (according to the second property of Definition 1.1). These comprise all the oracle queries made by our algorithm at time $t$. Therefore, the given bounds for this case hold. A detailed proof for the remaining bounds is provided in the appendix. □

## 3 Empirical results

In this section, we empirically study our $(8+\varepsilon)$-approximation dynamic algorithm. We implement our codes in C++ and run them on a MacBook laptop with 8 GB RAM and $M1$ processor. We empirically study the performance of our algorithm for video summarization and the Max-Cut problem.

**Video summarization.** Here, we use the Determinantal Point Process (DPP) which is introduced by [42], and combine it with our algorithm to capture a video summarization. We run our experiments on YouTube and Open Video Project (OVP) datasets from [20].

For each video, we use the linear method of [33] to extract a subset of frames and find a positive semi-definite kernel $L$ with size $n \times n$ where $n$ is the number of extracted frames. Then, we try to find a subset $S$ of frames such that it maximizes $\frac{det(L_S)}{det(I+L)}$ where $L_S$ is the sub-matrix of $L$ restricted to indices corresponding to frames $S$. Since $L$ is a positive semi-definite matrix, we have $det(L_S) \geq 0$. Interestingly, [39] showed that $\log(det(L_S))$ is a non-monotone function. We use these properties and set $f(S) := \log(det(L_S) + 1)$ to make $f$ a non-monotone non-negative submodular function. Then we run our $(8 + \varepsilon)$-approximate dynamic algorithm to find the best $S$ to maximize $f(S)$ such that $|S| \leq k$ for $k \in [10]$.

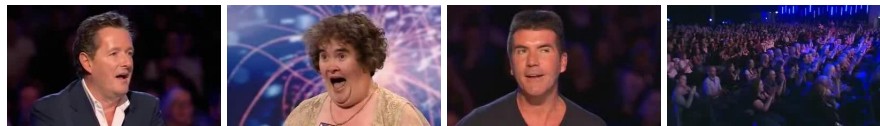

Figure 1: Video summarization of Susan Boyle's performance on Britain's Got Talent show (video 106) from YouTube.

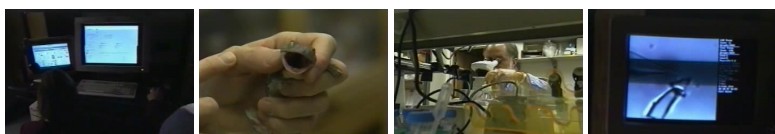

Figure 2: Video summarization for "Senses And Sensitivity, Introduct. to Lecture 1 presenter" (video 36) from OVP.

First, we insert all frames to observe the quality of our algorithm. Figure 1 and 2 are the selected frames by our algorithm for Video 106 from YouTube and Video 36 from OVP, respectively, when we limit the number of selected frames to $4$. Then, we create a sequence $\Xi$ of updates of frames of each video. Similar to [40], we define the sequence as a sliding window model. That is, given a window of size $W$ for a parameter $W \in \mathbb{N}$, a frame is inserted at a time $t$ and will be alive for a window of size $W$ and then we delete that frame.

To evaluate the performance of our algorithm, we benchmark (See Figure 3) the total number of query calls and the submodular value of set $S$ of our algorithm and the streaming algorithm proposed for non-monotone submodular maximization so-called SAMPLE-STREAMING proposed in [28]. This algorithm works as follows: Upon arrival of an element $u$, with probability $(1-q)$, for a parameter $0 < q < 1$, we ignore $u$, otherwise (i.e., with probability $q$), we do the following. If the size of set $S$ that we maintain is less than $k$, i.e, $|S| < k$ and $\Delta(u|S) > 0$, we add $u$ to $S$. However, if $|S| = k$, we select an element $v \in S$ for which $\Delta(v : S)$ is minimum possible, where $\Delta(u : S)$ equals to $\Delta(u|S_u)$ where $S_u$ are elements that arrived before $u$ in sequence $\Xi$. If $\Delta(u|S) \geq (1+c)\Delta(v : S)$ for a constant $c$, we replace $v$ by $u$; otherwise, we do nothing. Now we convert this streaming algorithm into a dynamic algorithm. To accomplish this, we restart SAMPLE-STREAMING after every deletion that deletes an element of solution set $S$ that is reported by SAMPLE-STREAMING's outputs. That is, if a deletion does not touch any element in set $S$, we do nothing; otherwise we restart the streaming algorithm.

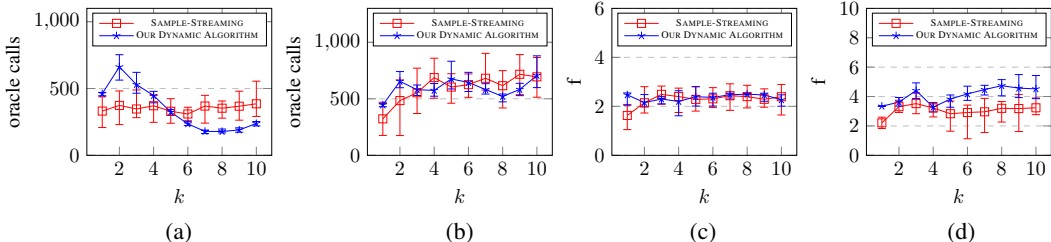

(a)  (b)  (c)  (d)

Figure 3: We plot the total number of query calls and the average output of our dynamic algorithm and SAMPLE-STREAMING on video 106 from YouTube and video 36 from OVP. In this figure, from left to right, Sub-figures (a) and (b) are the total oracle calls for video 106 and 36, respectively. Similarly, Sub-figures (c) and (d) are average submodular value for video 106 and 36, respectively.

We run our algorithm for $\varepsilon = k/2$ and compare the total oracle calls and average output of our algorithm and SAMPLE-STREAMING in Figure 3. To prove the approximation guarantee of our dynamic algorithm, we assumed $\varepsilon \leq 1$. However, in practice, it is possible to increase $\varepsilon$ up to a certain level without affecting the output of the algorithm significantly. On the other hand, increasing $\varepsilon$ reduces the total oracle calls and makes the algorithm faster. As you can see in Figure 3 plots (b) and (d), the submodular value of our algorithm is not worse than the SAMPLE-STREAMING algorithm whose approximation factor is $3 + 2\sqrt{2} \approx 5.828$ which is better than our approximation factor. Thus, our algorithm has an outcome better than our expectation, while its total oracle calls are better than SAMPLE-STREAMING algorithm (look at Figure 3 plots (a) and (c)).

We also empirically study the celebrated Max-Cut problem which is a non-monotone submodular maximization function (See [27]). These experiments are given in the appendix.

## 4 Conclusion

In this paper, we studied non-monotone submodular maximization subject to cardinality constraint $k$ in the dynamic setting by providing a reduction from this problem to maximizing monotone submodular functions under the cardinality constraint $k$ with a certain kind of algorithms($\tau$-thresholding algorithms). Moreover, we used our reduction to develop the first dynamic algorithms for this problem. In particular, both our algorithms maintain a solution set whose submodular value is a $(8 + \varepsilon)$-approximation of the optimal value and require $O(\varepsilon^{-3}k^3 \log^3(n) \log(k))$ and $O(\varepsilon^{-1}k^2 \log^3(k))$ oracle queries per update, respectively.

## 5 Acknowledgements

This work is partially supported by DARPA QuICC NSF AF:Small #2218678, and NSF AF:Small #2114269.

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
