# OpenReview forum: "Dynamic Non-monotone Submodular Maximization"
_NeurIPS.cc/2023/Conference — NeurIPS 2023 poster_

### Official Review · Reviewer_dvm4 · 2023-06-18

**Soundness:** 3 good
**Presentation:** 4 excellent
**Contribution:** 4 excellent
**Rating:** 6
**Confidence:** 4

**Summary:**

This work studies non-monotone submodular maximization subject to a cardinality
constraint in a fully dynamic setting, i.e., maintaining a good solution as
elements are inserted and deleted from the "current" ground set. Studying
non-monotone submodular maximization in this model is the natural follow-up to
the works of:
- Monemizadeh (NeurIPS 2020): $(0.5 - \varepsilon)$-approximation for *monotone* dynamic submodular maximization with amortized update time $O(k^2 \varepsilon^{-3} \log^{5} n)$.
- Lattanzi et al. (NeurIPS 2020): $(0.5 - \varepsilon)$-approximation for *monotone* dynamic submodular maximization with amortized update time $O(\log^4(k) \log^2 (n) / \varepsilon^{7})$.
- Chen and Peng (STOC 2020): proves query complexity lower bounds for approximation ratios strictly greater than $0.5$ for monotone dynamic submodular maximization. Gives $(1-1/e-\varepsilon)$-approximation for "insertion-only" streams for matroid constraints.

This work gives a $(1/8-\varepsilon)$-approximation for the *non-monotone*
fully dynamic cardinality-constrained problem with amortized query complexity
$O(\varepsilon^{-3} k^3 \log^3 n \log k)$ queries per update. The proposed
algorithm builds on connections to thresholding algorithms for *monotone*
submodular maximization and then altering these solutions to get guarantees for
non-monotone functions. The authors include experiments and compare their
algorithm to the Simple-Streaming in Feldman-Karbasi-Kazemi (NeurIPS 2018).

**Strengths:**

- Gives $(1/8 - \varepsilon)$-approximation for fully-dynamic non-monotone
  submodular maximization subject to a cardinality constraint, answering an
  open question in Chen-Peng (STOC 2020).
- Builds on thresholding techniques commonly used for monotone submodular
  maximization. This helps connect the toolkits for each problem type. Some
  missing references on line 54 when discussing thresholding:
  1. "Submodular Optimization in the MapReduce Model" (Liu-Vondrak, SOSA 2019)
  2. "Submodular Maximization with Nearly Optimal Approximation, Adaptivity and Query Complexity" (Fahrbach-Mirrokni-Zadimoghaddam, SODA 2019)
  3. "Fully Dynamic Algorithm for Constrained Submodular Optimization" (Lattanzi et al., NeurIPS 2020)
  4. "Practical and Parallelizable Algorithms for Non-Monotone Submodular Maximization with Size Constraint" (Chen-Kuhnle, arXiv:2009.01947, 2022)


**Weaknesses:**

- The biggest weakness of this paper are the experiments. They almost check all
  of the boxes, but they aren't very motivating. This paper investigates:
  1. video frame summary on the entire set, which doesn't use any aspect of
     streaming but is a reasonable starting place to show what the algorithms
     output.
  2. sliding window model of length $W$, which is fully dynamic but not in the
     most interesting way (though possibly the most practical way).
  3. only compares against the Sample-Streaming algorithm of
     Feldman-Karbasi-Kazemi (NeurIPS 2018). It would be nice to include
     comparisons to the three papers discussed in the abstract, too, even though
     they are for monotone submodular functions.
- Given that there is randomness in this paper's Update algorithm (line 4 of
  SubsetSelection), it is important for these experiments to be averaged over
  several trials with standard deviation error bars. Both the oracle calls and
  objective value plots appear somewhat noisy and non-monotone.


**Questions:**

**Questions**
- [line  74] This is less of a question and more of a comment:
  The authors are correct in questioning whether thresholding works in a
  non-monotone setting when a randomly sampled set is added to the current
  solution, i.e., $f(S_t) \ge k \tau$. See Chen-Kuhnle (2022) for an example
  where this property does not hold. However, also see Chen-Kuhnle (2022) and
  Fahrbach-Mirrokni-Zadimoghaddam (ICML 2019, arXiv:1808.06932v3) for a method
  to circumvent this problem.

**Typos and suggestions**
- [line 27] Two relevant missing works for non-monotone submodular maximization:
  1. "Non-monotone Submodular Maximization with Nearly Optimal Adaptivity and Query Complexity" (Fahrbach-Mirrokni-Zadimoghaddam, ICML 2019)
  2. "Practical and Parallelizable Algorithms for Non-Monotone Submodular Maximization with Size Constraint" (Chen-Kuhnle, arXiv:2009.01947, 2022)
- [line  38] suggestion: The description of the dynamic setting "... set of
  elements that are inserted but not deleted after their last insertion time
  till time $t$" is not clear and should be improved. The description in Section
  1.1 is better, but could still be improved (i.e., saying that $V_t$ is the set
  of ``active'' elements at time $t$).
- [line  71] typo: "The only such a result" --> "The only such result"
- [line 184] suggestion: Consider using $\text{Insert}_{i}(v, \tau)$ to avoid
  a double subscript.
- [line 313] typo in Figure 3 description: "submdoular"

---

> ### Author Rebuttal · Authors · 2023-08-10
>
> Thank you for your valuable comments. We will make sure to fix the issues you pointed out and incorporate your suggestions in the revised version of the paper.

---

> > ### Comment · Reviewer_dvm4 · 2023-08-13
> >
> > I read all the reviews and author rebuttals, and will keep my rating the same.
> >
> > I would like confirmation from the authors: Will you add standard deviation error bars to your plots in Figure 3? Better yet, would you include them in a one-page pdf response if the upload option is still available?

---

> > > ### Author Response · Authors · 2023-08-14
> > >
> > > Sure, as the confirmation, the latex file of our experiments, including plots of error bars are given below. In case you had any difficulty in running the latex file, we can provide a link to access the plots. Unfortunately, the NeurIPS guidelines prohibit link sharing, but if there is any other way that we can provide the links to plots, we will be happy to provide that.  In addition, we would like to mention our previous experiments that we did for the max-cut problem along with error bars were given in Appendix E.1 (see Figure 4).
> > >
> > > \begin{figure*}[h]
> > > \centering
> > > \begin{tabular}{@{}c@{}}
> > > \begin{tikzpicture}
> > > \pgfplotsset{width=8cm,compat=1.9}
> > > \begin{axis}[
> > >     xlabel={$k$},
> > >     ylabel={oracle calls},
> > >     ymin=0, ymax=1000,
> > >     xtick={2, 4, 6, 8, 10},
> > >     legend pos=north west,
> > >     ymajorgrids=true,
> > >     grid style=dashed,
> > >     legend style={nodes={scale=0.5, transform shape}},
> > > ]
> > >
> > > \addplot[
> > >     color=red,
> > >     mark=square,
> > >     error bars/.cd,
> > >     y dir=both,
> > >     y explicit,
> > >     error bar style={
> > >     color=red}
> > >     ]
> > >     coordinates {
> > >         (1, 331.20000) += (0, 106.80000) -= (0, 123.20000) (2, 373.60000) += (0, 108.40000) -= (0, 143.60000) (3, 346.80000) += (0, 142.20000) -= (0, 64.80000) (4, 371.00000) += (0, 77.00000) -= (0, 124.00000) (5, 329.80000) += (0, 94.20000) -= (0, 89.80000) (6, 309.50000) += (0, 50.50000) -= (0, 50.50000) (7, 368.80000) += (0, 80.20000) -= (0, 127.80000) (8, 352.70000) += (0, 54.30000) -= (0, 74.70000) (9, 367.60000) += (0, 112.40000) -= (0, 104.60000) (10, 385.60000) += (0, 169.40000) -= (0, 94.60000)
> > >     };
> > >     \addlegendentry{\textsc{Sample-Streaming}}
> > >
> > > \addplot[
> > >     color=blue,
> > >     mark=star,
> > >     error bars/.cd,
> > >     y dir=both,
> > >     y explicit,
> > >     error bar style={
> > >     color=blue}
> > >     ]
> > >     coordinates {
> > >         (1, 461.40000) += (0, 9.60000) -= (0, 15.40000) (2, 662.30000) += (0, 90.70000) -= (0, 99.30000) (3, 529.50000) += (0, 91.50000) -= (0, 64.50000) (4, 447.80000) += (0, 31.20000) -= (0, 54.80000) (5, 323.90000) += (0, 20.10000) -= (0, 24.90000) (6, 235.40000) += (0, 16.60000) -= (0, 12.40000) (7, 179.20000) += (0, 9.80000) -= (0, 16.20000) (8, 178.50000) += (0, 18.50000) -= (0, 10.50000) (9, 189.80000) += (0, 16.20000) -= (0, 16.80000) (10, 238.30000) += (0, 14.70000) -= (0, 18.30000)
> > >     };
> > >     \addlegendentry{\textsc{Our Dynamic Algorithm}}
> > > \end{axis}
> > > \end{tikzpicture}
> > >
> > > \begin{tikzpicture}
> > > \pgfplotsset{width=8cm,compat=1.9}
> > > \begin{axis}[
> > >     xlabel={$k$},
> > >     ylabel={f},
> > >     ymin=0, ymax=8,
> > >     xtick={2, 4, 6, 8, 10},
> > >     legend pos=north west,
> > >     ymajorgrids=true,
> > >     grid style=dashed,
> > >     legend style={nodes={scale=0.5, transform shape}},
> > > ]
> > >
> > > \addplot[
> > >     color=red,
> > >     mark=square,
> > >     error bars/.cd,
> > >     y dir=both,
> > >     y explicit,
> > >     error bar style={
> > >     color=red}
> > >     ]
> > >     coordinates {
> > >         (1, 1.62818) += (0, 0.44989) -= (0, 0.57743) (2, 2.14678) += (0, 0.65564) -= (0, 0.41891) (3, 2.48939) += (0, 0.33780) -= (0, 0.40127) (4, 2.39747) += (0, 0.34600) -= (0, 0.64868) (5, 2.28432) += (0, 0.52396) -= (0, 0.47845) (6, 2.30636) += (0, 0.45690) -= (0, 0.35722) (7, 2.43273) += (0, 0.48661) -= (0, 0.59661) (8, 2.39775) += (0, 0.46000) -= (0, 0.46214) (9, 2.28745) += (0, 0.42656) -= (0, 0.32351) (10, 2.39145) += (0, 0.50267) -= (0, 0.74555)
> > >     };
> > >     \addlegendentry{\textsc{Sample-Streaming}}
> > >
> > > \addplot[
> > >     color=blue,
> > >     mark=star,
> > >     error bars/.cd,
> > >     y dir=both,
> > >     y explicit,
> > >     error bar style={
> > >     color=blue}
> > >     ]
> > >     coordinates {
> > >         (1, 2.46816) += (0, 0.07933) -= (0, 0.42347) (2, 2.15307) += (0, 0.32738) -= (0, 0.17542) (3, 2.29888) += (0, 0.09777) -= (0, 0.19832) (4, 2.19944) += (0, 0.43743) -= (0, 0.58491) (5, 2.39385) += (0, 0.40503) -= (0, 0.38268) (6, 2.36480) += (0, 0.17151) -= (0, 0.35363) (7, 2.47486) += (0, 0.06145) -= (0, 0.17877) (8, 2.49553) += (0, 0.03520) -= (0, 0.02626) (9, 2.46536) += (0, 0.07095) -= (0, 0.25866) (10, 2.26592) += (0, 0.20335) -= (0, 0.26033)
> > >     };
> > >     \addlegendentry{\textsc{Our Dynamic Algorithm}}
> > > \end{axis}
> > > \end{tikzpicture}
> > > \end{tabular}
> > >
> > > \small (a) Video 106 total oracle calls and average output

---

> > > > ### Author Response · Authors · 2023-08-14
> > > > **Continuation of latex**
> > > >
> > > > \begin{tabular}{@{}c@{}}
> > > > \begin{tikzpicture}
> > > > \pgfplotsset{width=8cm,compat=1.9}
> > > > \begin{axis}[
> > > >     xlabel={$k$},
> > > >     ylabel={oracle calls},
> > > >     ymin=0, ymax=1300,
> > > >      xtick={2, 4, 6, 8, 10},
> > > >     legend pos=north west,
> > > >     ymajorgrids=true,
> > > >     grid style=dashed,
> > > >     legend style={nodes={scale=0.5, transform shape}},
> > > > ]
> > > >
> > > > \addplot[
> > > >     color=red,
> > > >     mark=square,
> > > >     error bars/.cd,
> > > >     y dir=both,
> > > >     y explicit,
> > > >     error bar style={
> > > >     color=red}
> > > >     ]
> > > >     coordinates {
> > > >         (1, 322.70000) += (0, 146.30000) -= (0, 146.70000) (2, 484.00000) += (0, 183.00000) -= (0, 309.00000) (3, 553.50000) += (0, 217.50000) -= (0, 183.50000) (4, 687.60000) += (0, 171.40000) -= (0, 182.60000) (5, 603.70000) += (0, 118.30000) -= (0, 142.70000) (6, 625.30000) += (0, 94.70000) -= (0, 112.30000) (7, 681.60000) += (0, 220.40000) -= (0, 104.60000) (8, 615.80000) += (0, 133.20000) -= (0, 196.80000) (9, 717.10000) += (0, 173.90000) -= (0, 180.10000) (10, 694.10000) += (0, 170.90000) -= (0, 180.10000)
> > > >     };
> > > >     \addlegendentry{\textsc{Sample-Streaming}}
> > > >
> > > > \addplot[
> > > >     color=blue,
> > > >     mark=star,
> > > >     error bars/.cd,
> > > >     y dir=both,
> > > >     y explicit,
> > > >     error bar style={
> > > >     color=blue}
> > > >     ]
> > > >     coordinates {
> > > >         (1, 445.80000) += (0, 14.20000) -= (0, 15.80000) (2, 654.50000) += (0, 86.50000) -= (0, 56.50000) (3, 580.50000) += (0, 41.50000) -= (0, 51.50000) (4, 574.30000) += (0, 60.70000) -= (0, 51.30000) (5, 675.40000) += (0, 156.60000) -= (0, 88.40000) (6, 644.90000) += (0, 89.10000) -= (0, 79.90000) (7, 581.80000) += (0, 52.20000) -= (0, 39.80000) (8, 520.90000) += (0, 21.10000) -= (0, 47.90000) (9, 583.10000) += (0, 53.90000) -= (0, 55.10000) (10, 706.50000) += (0, 173.50000) -= (0, 107.50000)
> > > >     };
> > > >     \addlegendentry{\textsc{Our Dynamic Algorithm}}
> > > > \end{axis}
> > > > \end{tikzpicture}
> > > >
> > > > \begin{tikzpicture}
> > > > \pgfplotsset{width=8cm,compat=1.9}
> > > > \begin{axis}[
> > > >     xlabel={$k$},
> > > >     ylabel={f},
> > > >     ymin=0, ymax=10,
> > > >     xtick={2, 4, 6, 8, 10},
> > > >     legend pos=north west,
> > > >     ymajorgrids=true,
> > > >     grid style=dashed,
> > > >     legend style={nodes={scale=0.5, transform shape}},
> > > > ]
> > > >
> > > > \addplot[
> > > >     color=red,
> > > >     mark=square,
> > > >     error bars/.cd,
> > > >     y dir=both,
> > > >     y explicit,
> > > >     error bar style={
> > > >     color=red}
> > > >     ]
> > > >     coordinates {
> > > >         (1, 2.20679) += (0, 0.38838) -= (0, 0.38206) (2, 3.30840) += (0, 0.44028) -= (0, 0.39134) (3, 3.51886) += (0, 0.73748) -= (0, 0.68497) (4, 3.22474) += (0, 0.36300) -= (0, 0.61065) (5, 2.83012) += (0, 0.86126) -= (0, 1.18679) (6, 2.91402) += (0, 0.50340) -= (0, 1.78629) (7, 2.96153) += (0, 0.91239) -= (0, 1.41191) (8, 3.19881) += (0, 0.46492) -= (0, 0.93366) (9, 3.18183) += (0, 0.96342) -= (0, 1.56121) (10, 3.23328) += (0, 0.66669) -= (0, 0.46605)
> > > >     };
> > > >     \addlegendentry{\textsc{Sample-Streaming}}
> > > >
> > > > \addplot[
> > > >     color=blue,
> > > >     mark=star,
> > > >     error bars/.cd,
> > > >     y dir=both,
> > > >     y explicit,
> > > >     error bar style={
> > > >     color=blue}
> > > >     ]
> > > >     coordinates {
> > > >         (1, 3.34082) += (0, 0.00204) -= (0, 0.00613) (2, 3.62122) += (0, 0.31347) -= (0, 0.27428) (3, 4.40612) += (0, 0.52449) -= (0, 0.53673) (4, 3.27633) += (0, 0.29102) -= (0, 0.32123) (5, 3.82531) += (0, 0.28081) -= (0, 0.55184) (6, 4.18082) += (0, 0.52530) -= (0, 0.44204) (7, 4.47143) += (0, 0.29184) -= (0, 0.36531) (8, 4.73959) += (0, 0.41143) -= (0, 0.69061) (9, 4.56939) += (0, 0.92449) -= (0, 0.63878) (10, 4.52245) += (0, 0.91020) -= (0, 0.66531)
> > > >     };
> > > >     \addlegendentry{\textsc{Our Dynamic Algorithm}}
> > > > \end{axis}
> > > > \end{tikzpicture}
> > > > \end{tabular}
> > > >
> > > > \small (b) Video 36 total oracle calls and average output
> > > >
> > > >
> > > > \caption{We plot the total number of query calls and the average output of our dynamic algorithm and \textsc{Sample-Streaming}
> > > > on video 106 from YouTube and video 36 from OVP.}
> > > > \end{figure*}

---

### Official Review · Reviewer_PoYA · 2023-06-27

**Soundness:** 3 good
**Presentation:** 3 good
**Contribution:** 2 fair
**Rating:** 5
**Confidence:** 4

**Summary:**

In this paper, the authors consider the non-monotone submodular maximization problem under the cardinality constraint and dynamic model. Here, the dynamic model means that the ground set of the submodular function changes every time step where one element is inserted into or deleted from the ground set, and the update is controlled by an oblivious adversarial. They show a reduction from non-monotone case to monotone case, and then obtain a dynamic algorithm with (8+eps)-approximation ratio. They further test their algorithm on some real-world data sets.

**Strengths:**

1. In the dynamic setting, they provide a constant approximation algorithm for non-monotone submodular maximization problem under cardinality constraint.
2. The reduction between non-monotone dynamic algorithm and monotone dynamic algorithm shows some connection between these two cases.

**Weaknesses:**

1. I’m confused with the relation between parameter $\tau$ and the results in theorem 1.2. The current description looks like there is no relationship between these two. But, if $\tau$ is large enough, it seems we can satisfy definition 1.1 by keeping $S_t$ an empty set. For example, we can assume without loss of generality $f(u)\leq 1$ for any u and set $\tau =1$. In this case, we have a trivial $\tau$-thresholding dynamic algorithm. Then, what does theorem 1.2 look like?
2. Although the paper tries to show the connection between monotone case and non-monotone case by the reduction theorem, it seems that this connection is weak. Firstly, to show a monotone dynamic algorithm is \tau-thresholding algorithm is not an easy task, so that it is not easy to use this reduction. More importantly, the reduction does not mean that a better monotone dynamic algorithm can imply a better non-monotone dynamic algorithm (comparing to the reduction result in [35]). The better approximation ratio of the monotone dynamic algorithm does not help. It is also not clear whether the lower amortized update time can help or not.

The author's response has clarified my main concern here, so I would like to raise my score.

**Questions:**

See weakness 1

**Limitations:**

Yes

---

> ### Author Rebuttal · Authors · 2023-08-10
>
> Thank you for your review. Please see below for our answers to your comments:
>
> >I’m confused with the relation between parameter $\tau$ and the results in theorem ...
>
> Thank you for pointing out this issue. It appears that the writing here has been confusing, and you may have misunderstood the theorem statement.
> The assumption of the theorem is that there exists a $\tau$-thresholding for **any** given value of $\tau$. In other words, $\tau$ is an input parameter. Therefore, one cannot simply set $\tau=1$ and obtain an algorithm for this value; the algorithm should work for *any* value of tau.
> Specifically, our reduction sets $\tau = \frac{OPT}{k(3+1/(2\alpha))}$, where OPT denotes our guess for the optimal value.
>
> Please let us know if you have any additional questions regarding the theorem.
>
> >Although the paper tries to show the connection between monotone case and non-monotone case by ...
>
> We answer different parts of your comment separately to cover all the points you made, with the hope of resolving all of your concerns.
>
> Firstly, regarding your concern about the difficulty of showing that an algorithm is $\tau$-thresholding:
> We agree that for any monotone algorithm, the $\tau$-thresholding property would need to be formally verified (e.g., we do this in our paper in Appendix D).
> However, all dynamic algorithms known for monotone submodular maximization under cardinality constraint $k$ have been based on the thresholding algorithm proposed in [1], and it is not unreasonable to expect current and even future dynamic algorithms to be $\tau$-thresholding.
>
> Secondly, regarding your concern about improvements in monotone dynamic algorithms:
> Chen and Peng in [2] show that $\frac{1}{2}$ is a tight approximation guarantee for the dynamic submodular maximization problem. Therefore, we do not expect any improvement in the approximation guarantee of monotone dynamic algorithms. However, the query complexity of dynamic algorithms for the problem of monotone submodular maximization under cardinality constraint can be improved. As we observe in Theorem 1.2, if such improvements are obtained using a $\tau$-thresholding algorithm, this will improve the update time (measured by query complexity) of our dynamic algorithm for the non-monotone version as well.
>
> It is worth mentioning that a new dynamic algorithm for monotone submodular maximization has been proposed in [3] (also mentioned by another reviewer). We believe it can be shown that this new and improved algorithm is also a $\tau$-thresholding algorithm, which works fine in our reduction, and our reduction has already resulted in a better dynamic algorithm for non-monotone submodular maximization under cardinality constraint $k$.
>
> Lastly, regarding the reduction in [35] ([4] below), we should mention that their reduction does not seem to be general as well. In their paper, they claim that the solution obtained by approximation algorithms for monotone submodular functions often satisfies $f(S) \geq \alpha f(S \cup C^∗)$, where $1 \geq \alpha > 0$, and $C^∗$ is the optimal solution. (See Section Streaming Local Search for a collection of independence systems on page 3.) They use this as an assumption in their proof for their Theorem 1. (See equation (2) in proof of theorem 1 in supplementary materials). Hence, we believe their reduction also only works for monotone algorithms with that particular property and not any general algorithm.
>
>
> [1] Ashwinkumar Badanidiyuru, Baharan Mirzasoleiman, Amin Karbasi, Andreas Krause:
> Streaming submodular maximization: massive data summarization on the fly. KDD 2014: 671-680
>
> [2] Xi Chen, Binghui Peng: On the complexity of dynamic submodular maximization. STOC 2022: 1685-1698
>
> [3] Kiarash Banihashem, Leyla Biabani, Samira Goudarzi, MohammadTaghi Hajiaghayi, Peyman Jabbarzade, Morteza Monemizadeh: Dynamic Algorithms for Matroid Submodular Maximization. CoRR abs/2306.00959 (2023)
>
> [4] Baharan Mirzasoleiman, Stefanie Jegelka, Andreas Krause:
> Streaming Non-monotone Submodular Maximization: Personalized Video Summarization on the Fly. http://arxiv.org/abs/1706.03583

---

> > ### Author Response · Authors · 2023-08-20
> >
> > Dear reviewer,
> >
> > We were wondering whether our response has addressed your concerns, especially regarding the statement of Theorem 1.
> > We'd be happy to provide any additional clarifications.

---

> > > ### Comment · Reviewer_PoYA · 2023-08-21
> > >
> > > Thanks for the explaination of my concern. I think the response has already addressed my main concern and I would like to raise the score.

---

### Official Review · Reviewer_4fuM · 2023-07-04

**Soundness:** 3 good
**Presentation:** 3 good
**Contribution:** 4 excellent
**Rating:** 7
**Confidence:** 4

**Summary:**

The paper studies the dynamic submodular maximization problem and gives the first efficient dynamic algorithm that maintains a constant approximation solution.

In the problem of dynamic submodular maximization, there is a sequence of updates (insertions/deletions) of ground set element, and one wants to maintain a subset (of size at most $k$) that obtains the maximum possible function value. Previous work provide tight approximation guarantee when the function is monotone, while this paper extends to non-monotone submodular function (albeit with worse approximation guarantee).

The main result of this paper is an $(8+\epsilon)$ approximation algorithm with $\mathsf{poly}(k, \log n, \epsilon^{-1})$ amortized update time per update. The algorithm is obtained through a clever reduction from dynamic algorithms for monotone function (and satisfies certain threshold property). The overall idea is to simulate the monotone threshold algorithm twice and combines double greedy or random sampling. The first run guarantees the marginal is small for the second algorithm, if it works bad.

Besides theoretical results, the author also conduct empirical study and verify its practicality.


--------------
I have read the author's response and I keep my positive evaluation of the paper.



**Strengths:**

The paper provides the first sublinear dynamic algorithm for non-montonne submodular maximization. The theoretical contribution is novel and interesting from my perspective. The experiment result is also promising.

**Weaknesses:**

There is no major weakness. Some minor issues are listed below.


(1) Line 201, it seems one can directly use the double greedy algorithm to extract $S_1'$, I don't see why you mention the 3-approximation deterministic algorithm.

(2) Line 242, delete "what"

(3) The related work is quite extensive, there are a few missing reference that are relevant to the paper. The first two reference below seems appear on Arxiv later than NeurIPS submission, but I encourage the author to add the reference in the later version. The third reference gives lower bound for white box dynamic submodular maximization.

[1] Dynamic Algorithms for Matroid Submodular Maximization. Kiarash Banihashem, Leyla Biabani, Samira Goudarzi, MohammadTaghi Hajiaghayi, Peyman Jabbarzade, Morteza Monemizadeh

[2] Fully Dynamic Submodular Maximization over Matroids. ICML 2023
Paul Dütting, Federico Fusco, Silvio Lattanzi, Ashkan Norouzi-Fard, Morteza Zadimoghaddam

[3] Dynamic influence maximization. NeurIPS 2021
Binghui Peng







**Questions:**

.

**Limitations:**

.

---

> ### Author Rebuttal · Authors · 2023-08-10
>
> Thank you for pointing out the issues and listing the extra references. We will make sure to incorporate them in the revised version of the paper.

---

### Official Review · Reviewer_Ma6Y · 2023-07-06

**Soundness:** 4 excellent
**Presentation:** 2 fair
**Contribution:** 3 good
**Rating:** 5
**Confidence:** 4

**Summary:**

The authors consider an submodular non-monotone optimization problem in fully dynamic setting. The authors propose a 8+epsilon approximation algorithm by combining several existing methods.

**Strengths:**

- The technique behind the proposed algorithm is interesting.
- Proven approximation guarantee.
- A nice continuation of the existing literature in submodular optimization.

**Weaknesses:**

- 8 + epsilon is a rather weak approximation guarantee.
- The other algorithm with 10 + epsilon approximation uses a weak algorithm, so the algorithm is largely pedagogical.
- Presentation needs improvement.

**Questions:**

The authors should improve the presentation of the paper

- The abstract is probably better without references to Neurips or STOC
- Line 20-22. Provide citations.
- Page 2: remove grey outline boxes.
- Line 51, remove bolding
- Property 2. queries is not defined.
- Reword/remove metatheorem.
- Line 98: . That is -> , that is,
- Line 178: algorithms -> algorithm
- Named paragraphs starting on Page 5 need connecting text.
- There is a follow-up paper of [7] that derandomizes the algorithm of [7] (and keeping 2-approximation). Can this algorithm used?

**Limitations:**

Probably the biggest limitation of the method is a rather weak approximation guarantee. It is not clear whether a stronger approximation is possible. This is not discussed by the authors.

---

> ### Author Rebuttal · Authors · 2023-08-09
>
> Thank you for your review. Please see below for our answers to your comments:
>
> > 8 + epsilon is a rather weak approximation guarantee.
>
> Our primary goal in this paper was to obtain the first dynamic constant-factor approximation algorithm for non-monotone submodular maximization. This answers affirmatively an open question posed by Chen and Peng in [11]. Indeed, developing a dynamic algorithm for this problem with a better approximation factor than $(8 +\epsilon)$ is an interesting future direction in this area.
> Yet, it is worth mentioning that non-monotone submodular maximization is far more difficult than monotone submodular maximization. Monotone submodular maximization under cardinality constraint has a tight $\frac{e}{e -1}$ approximation algorithm in the offline mode and nearly tight $2 + \epsilon$ approximation algorithms for both streaming and dynamic settings. However, for the non-monotone version, there is a hardness result that says it is impossible to obtain a $2.04$ (i.e., $0.491$) approximation algorithm for this problem even in the offline setting, and to the best of our knowledge, the current state of the art algorithms for this problem have $2.6$ and $3.6$ (i.e., $0.385$ and $0.2779$) approximation guarantee for offline and streaming settings respectively. Hence, we believe our algorithm with $(8+\epsilon)$ approximation guarantee as the first non-monotone algorithm in the dynamic setting is valuable.
>
> > Probably the biggest limitation of the method is a rather weak approximation guarantee. It is not clear whether a stronger approximation is possible. This is not discussed by the authors.
>
> The reduction resulting in the $(10 + \epsilon)$-approximation algorithm is interesting because of its potential to develop a dynamic algorithm with polylog(n) oracle queries for non-monotone submodular maximization. In particular, if one can show that a dynamic algorithm for monotone submodular maximization with polylog(n) oracle queries is a $\tau$-thresholding dynamic algorithm, our reduction immediately provides a counterpart dynamic algorithm for non-monotone submodular maximization with polylog(n) oracle queries.
>
> > The abstract is probably better without ...
>
> Thank you for your comments on the presentation. We will incorporate them in the revised version of the paper.
>
> > There is a follow-up paper of [7] that derandomizes the algorithm of [7] (and keeping 2-approximation). Can this algorithm be used?
>
> The dynamic algorithm from [38] that we use as our $\tau$-thresholding algorithm is a randomized algorithm. Therefore, even if we use the derandomized version of the algorithm of [7], our dynamic algorithm would remain randomized.
>
> > Probably the biggest limitation of the method is a rather weak approximation guarantee. It is not clear whether a stronger approximation is possible. This is not discussed by the authors.
>
> Please see our response to the first comment.

---

### Decision · Program_Chairs · 2023-09-21

**Decision:**

Accept (poster)

**Comment:**

This paper gives the first constant factor approximation algorithm for non-monotone submodular maximization under a cardinality constraint in the fully dynamic setting with sublinear update time. Dynamic submodular maximization is a recent and active area of research in submodular optimization and this paper makes substantial progress in this area with the first algorithm for non-monotone functions. The techniques are also interesting and novel. I recommend acceptance.